# Analysis of Long Noncoding RNA and mRNA Expression Profiles of Testes with High and Low Sperm Motility in Domestic Pigeons (*Columba livia*)

**DOI:** 10.3390/genes11040349

**Published:** 2020-03-25

**Authors:** Xiuli Xu, Yuge Tan, Haiguang Mao, Honghua Liu, Xinyang Dong, Zhaozheng Yin

**Affiliations:** Animal Science College, Zhejiang University, Zijingang Campus, Hangzhou 310058, China; 21817083@zju.edu.cn (X.X.); 21817066@zju.edu.cn (Y.T.); maohaiguang@163.com (H.M.); 21717075@zju.edu.cn (H.L.); sophiedxy@zju.edu.cn (X.D.)

**Keywords:** lncRNAs, mRNA, RNA-seq, sperm motility, testis, pigeon

## Abstract

Sperm motility is one of the most important indicators in assessing semen quality, and it is used to evaluate poultry fertility. Many long noncoding RNAs (lncRNAs) and mRNAs are involved in regulating testis development and spermatogenesis. In this study, we employed RNA sequencing to analyse the testis transcriptome (lncRNA and mRNA) of ten pigeons with high and low sperm motility. In total, 46,117 mRNAs and 17,463 lncRNAs were identified, of which 2673 mRNAs and 229 lncRNAs (*P* < 0.05) were significantly differentially expressed (DE) between the high and low sperm motility groups. Gene ontology (GO) and Kyoto Encyclopedia of Genes and Genomes (KEGG) annotation analysis showed that target genes of DE lncRNAs and DE mRNAs were related to calcium ion binding, ATP binding, and spermatogenesis. Moreover, we found that UBB, a target gene of lncRNA MSTRG.7787.5, was involved in germ cell development. Our study provided a catalogue of lncRNAs and mRNAs associated with sperm motility, and they deserve further study to deepen the understanding of biological processes in the pigeon testis.

## 1. Introduction

Semen quality is usually evaluated based on several characteristics, such as ejaculation volume, density, motility, and deformity rate. Semen quality has an important role in determining fertility, which is an important indicator in the economics of producing livestock. Motility is one of the basic functions of mature male gametes, and it is an important indicator of semen quality. Sperm mobility is a heritable quantitative trait, which indicates the ability of a sperm cell population to move against resistance at body temperature, which is considered to be positively related to fertility [1,2]. The testes are vital for maintaining male fertility, and their main functions are spermatogenesis and androgen biosynthesis. Many pathways and genes have been reported to be involved in sperm motility [3].

Long noncoding RNAs (lncRNAs) are a large class of noncoding RNAs longer than 200 bp, and they were once thought to be transcriptional noise. Many experiments have demonstrated that lncRNAs play critical roles in various aspects of gene regulation, such as epigenetic regulation, X chromosome inactivation, genomic imprinting, and chromatin modification [4,5]. Zhu et al. [6] found that the main function of lncRNAs is to bind to DNA, RNA and proteins to regulate gene expression. The expression of lncRNAs was found specifically in tissues, and many lncRNAs were preferentially expressed in the brain and testes [7]. LncRNAs have also been found to be key regulators in a wide range of biological processes, including reproduction [8]. Some lncRNAs have been identified to play vital roles in testis development and spermatogenesis [9,10]. A variety of testicular-specific lncRNAs have been detected in several species [11,12,13]. However, lncRNAs, as key regulatory factors in many biological processes, have not been systematically identified in the testes of pigeons in relation to different fertility rates.

We sought a better understanding of the relevant molecular regulatory mechanisms of sperm motility. In this study, we performed transcriptome analysis of ten pigeon testes with differential sperm motility using RNA sequencing (RNA-seq) for the first time. The purpose of this study was to uncover the potential role of lncRNAs in spermatogenesis and further provide new insight into molecular mechanisms involved in the sperm motility in birds. Our data provide a basis for understanding the functional effects of lncRNAs on improving sperm motility in pigeons.

## 2. Materials and Methods 

### 2.1. Ethical Statement

All experimental protocols involving animals were approved by the Animal Care and Welfare Committee of Animal Science College and the Scientific Ethical Committee of the Zhejiang University (No. ZJU14814) (Hangzhou, China).

### 2.2. Sampling

The birds used for RNA-seq were white feather king pigeons (Columba livia), which were provided by Huzhou Huzhou Huajia Special Culture Co. (Zhejiang, China). All the pigeons were housed under the same conditions in a windowed cage with a male and female pair in each cage. To identify pigeons with different sperm motility for sequencing, sperm motility was assessed 5 times in 100 male birds that were two years old using the microscopic method at 2-day intervals. Fresh semen was diluted and the diluted semen (1:100 in PBS) was used to assess sperm motility at 400* magnification under at least five microscopic fields. In each experiment, the percentage of motile sperm was evaluated three times by the same observer [14]. Sperm motility was expressed as the number of motile spermatozoa with moderate to rapid progressive movement on a 10-point scale. According to the measured sperm motility of each bird, the birds with the 5 highest (HS1, HS2, HS3, HS4 and HS5) and 5 lowest (LS1, LS2, LS3, LS4 and LS5) motility phenotypes were selected for use as candidates for RNA-seq. The sperm motility rates of these 10 birds are shown in Table 1.

Finally, all of the birds were euthanized for tissue sampling. Testicular samples were dissected, temporarily frozen in liquid nitrogen, and stored at −80 °C until further manipulation.

### 2.3. LncRNA Library Construction and Sequencing

Total RNA was isolated and purified from each testicular sample using TRIzol reagent (Invitrogen, Carlsbad, CA, USA) according to the manufacturer’s procedure. The RNA concentration and integrity of individual testis samples were estimated using a NanoDrop ND-1000 (NanoDrop, Wilmington, DE, USA), and an Agilent 2100 with RIN number >7.0, respectively.

Ribosomal RNA was depleted from approximately 5 µg of total RNA according to the instructions of the Ribo-Zero rRNA Removal kit (Illumina, San Diego, USA). The remaining RNAs were fragmented into small fragments using divalent cations. The cleaved RNA fragments were then reverse-transcribed to produce cDNA. The average size of cDNA fragments in the library was 300 bp (±50 bp). Finally, we performed paired-end sequencing on an Illumina HiSeq 4000 (LC Bio, China) following the vendor’s recommended protocol.

### 2.4. Quality Control and Mapping

First, Cutadapt was used to remove low-quality reads containing adaptor contamination, low quality bases, and undetermined bases. FastQC was then used to verify sequence quality. We used Bowtie2 [15] and Hisat2 [16] to map the reads to the pigeon’s genome in NCBI (ftp://ftp.ncbi.nlm.nih.gov/genomes/all/GCF/000/337/935/GCF_000337935.1_Cliv_1.0/). StringTie [17] was used to assemble mapped reads for each sample. Then, all transcripts from pigeon samples were combined to reconstruct a comprehensive transcriptome using a Perl script. After the final transcriptome was generated, the expression levels of all transcripts were assessed using StringTie [17] and Ballgown [18].

### 2.5. LncRNA Identification

First, the transcripts that overlapped with known mRNAs and those that were shorter than 200 bp in length were discarded. We then used CPC [19] and CNCI [20] to predict transcripts with coding potential. All transcripts with CPC scores < −1 and with CNCI scores < 0 were removed. The remaining transcripts were considered lncRNAs.

### 2.6. Different Expression Analysis of mRNAs and lncRNAs 

StringTie was used to determine the expression levels of mRNAs and lncRNAs by calculating the FPKM (fragments per kilobase of exon per million mapped fragments). The DE mRNAs and lncRNAs were identified as having a log_2_ (fold change) > 1 or log_2_ (fold change) < −1 and a statistical significance (*p* value < 0.05), as determined by R package-edgeR. 

### 2.7. Target Gene Prediction and Functional Analysis of LncRNAs 

To explore the function of lncRNAs, we predicted the cis-target genes of lncRNAs. LncRNAs may play a cis role acting on neighboring target genes. In this study, 100 kbp upstream and downstream of coding genes were selected through python script [21]. The function of lncRNA was analyzed by the co-expression analysis of DE lncRNA and DE genes. Pearson correlation analysis was used to examine the correlations between DE lncRNAs and mRNAs. Then, we demonstrated the functional analysis of the target genes of lncRNAs using BLAST2GO [22]. 

### 2.8. GO and KEGG Enrichment Analysis

To better understand the main functions of DE lncRNAs and mRNA between the high and low sperm motility groups, GO terms and KEGG pathway enrichment analysis was performed to investigate the biological processes.

### 2.9. qPCR Validation

For qPCR analysis we randomly selected 6 lncRNAs (MSTRG.32367.1, MSTRG.13392.1, MSTRG.22946.2, MSTRG.33664.2, MSTRG.17590.3 and MSTRG.16178.2) and 6 mRNAs (SPRTN, NFRKB, LOC102083535, KIF16B, TOMM70 and CLASP1) that represent differential expression levels of RNA-seq from the testicles of 10 individuals. qPCR was performed on a StepOnePlus Real-Time PCR System (Applied Biosystems, USA) using SYBR Green PCR Master Mix (TaKaRa, HangZhou, China). β-actin was used as a control, and the relative expression levels of genes and lncRNAs were calculated using the 2^-ΔΔCt^ method. All lncRNA and mRNA expression was examined with three independent biological replicates. The primers for mRNAs and lncRNAs are shown in Appendix A.

## 3. Results

### 3.1. RNA Sequencing and Identification of LncRNAs and mRNAs in Pigeon Testes

To identify DE lncRNAs and mRNAs related to pigeon reproduction, ten cDNA libraries were generated from five birds with high sperm motility and five birds with low sperm motility. The raw reads with an average of 100 million 150 bp paired ends were acquired by an Illumina HiSeq 4000 Platform. After quality control with the filtering out of reads containing adaptor contamination, low-quality bases, and undetermined bases, approximately 13.93 Gb of high-quality sequence data was obtained for each sample. The GC content was approximately 47%. More than 84% of clean reads were mapped to the pigeon genome using TopHat, including 64.54% uniquely mapped reads. These detailed data are shown in Appendix A.

CNCI and CPC tools were used to calculate the protein coding potency of transcripts in this study. According to the structural and noncoding potential characteristics of lncRNAs identified in our study, a total of 17,463 putative lncRNAs were identified from the ten libraries. Regarding the genomic locations of the lncRNAs, 3383 were intronic (19.37%), 3201 were bidirectional (18.33%), 1039 were sense (5.95%), 8438 were intergenic (48.32%), and 1402 were antisense lncRNAs (8.03%). Moreover, detailed information on the identified lncRNAs is listed in Appendix A. 

In this study, the average length of lncRNA transcripts was 2842 bp, which is shorter than the 4141 bp length of mRNA transcripts, thus indicating that lncRNAs were shorter than mRNAs (Figure 1A). Additionally, the number of exons in lncRNAs (1.04 on average) was less than that of mRNAs (8.41 on average). A total of 61.89% of mRNAs had five or more exons, while 89.09% of lncRNAs had three or fewer exons (Figure 1B). Moreover, lncRNAs tended to have shorter ORFs than mRNAs in the testis (Figure 1C,D).

### 3.2. Differentially Expressed mRNAs and LncRNAs

To investigate the differences between high and low sperm motility groups, we examined the DE lncRNAs and DE mRNAs with FPKM levels in pigeon testes. As shown in Figure 2A, the expression levels of lncRNAs were higher than those of mRNAs, while the number of lncRNAs was less than those of mRNAs (Figure 2B).

The DE lncRNAs and mRNAs between the high and low sperm motility groups were analysed using edgeR software with a set filter of |log2 (fold change)| > 1, and *p* < 0.05. As a result, 2673 mRNAs and 229 lncRNAs were found to be significantly differentially expressed between the high and low sperm motility pigeon testes. In the low sperm motility group, 102 lncRNAs and 1369 mRNAs were significantly upregulated, while 127 lncRNAs and 1304 mRNAs were downregulated in the high sperm motility group. A volcano plot displays DE lncRNAs and mRNAs (Figure 3). 

### 3.3. Enrichment Analysis of Differentially Expressed mRNAs

Gene ontology, which is the key functional classification adopted by GO (http://geneontology.org), was used to analyse the main functions of DE mRNAs. A total of 578 GO terms with functional annotation information were enriched for 2673 differentially expressed mRNAs. There were 194 GO terms significantly enriched in the GO results that met the criteria of *P* < 0.05 (Appendix A). The significantly enriched GO terms included immune response, regulation of cell proliferation, oocyte differentiation, sperm capacitation, extracellular exosome and structural molecule activity (Figure 4A,B). KEGG pathway analysis recognized 10 significantly enriched pathways (*P* < 0.05), such as cell adhesion molecules, protein digestion and absorption and steroid hormone biosynthesis (Figure 4C). Detailed information is shown in Appendix A.

### 3.4. Cis-Regulatory Roles of Differentially Expressed LncRNAs in Testes

To further explore the regulatory functions of lncRNAs, we predicted the cis-regulated target genes of DE lncRNAs between high and low sperm motility groups. As a result, 275 potential lncRNA target genes were found in this study when using 100 kbp as the cutoff. Based on those cis-regulated target genes, the results of GO analysis identified 48 significant GO terms (*p* < 0.05) (Appendix A). Based on biological process analysis, DE lncRNA target genes were found to be involved in cell cycle arrest, positive regulation of protein lipidation, ribosome biogenesis and apoptotic process. The cellular component and molecular function categories identified were mainly related to the oligosaccharyltransferase complex, the anchored component of membrane, and hormone activity (Figure 5A,B). KEGG pathway enrichment analysis results suggested that these lncRNA target genes were mainly involved in pyrimidine metabolism, the cytosolic DNA-sensing pathway, Epstein–Barr virus infection, glutamatergic synapse and lysine degradation (Figure 5C, Appendix A). 

Based on the predicted results of DE lncRNA-gene pairs in cis, the first five and the last three lncRNA-gene pairs were listed in Table 2 according to the Pearson Correlation Coefficient. As shown in Table 2, the regulation directions of the first five pairs of lncRNA-gene pairs were the same, while those of the last three pairs were opposite.

### 3.5. Co-Enriched GO Terms of DE LncRNA and mRNA 

In order to identify the key pathways for regulating sperm motility, we identified five significantly enriched GO terms in both the DE mRNA enrichment and the lncRNA target gene enrichment. As shown in Table 3, the significantly enriched GO terms were involved in the anchored component of membrane, hormone activity, calcium channel activity, mitotic spindle and cytoskeleton, of which three pathways were involved in the cellular component and the other two pathways involved in molecular function.

### 3.6. Verification of Differentially Expressed LncRNAs and mRNAs 

To validate the RNA-seq results, we randomly selected six DE mRNAs and six DE lncRNAs for RT-qPCR analysis. As shown in Figure 6, the expression results of qPCR were consistent with those of seq-RNA, indicating that the seq-RNA data were accurate and credible.

## 4. Discussion

Sperm motility is a reliable and objective reflection of, and thus measure of, semen quality [23]. Semen quality plays a vital role in fertility in poultry. Moreover, males play the dominant role in fertility rather than females in animals. Sperm motility has been recognized to be correlated with fertility [24,25]. Sperm motility is regulated by many critical genes and complex regulatory pathways. However, the mechanism underlying sperm motility in pigeons is unclear. Hence, in this study, we constructed 10 cDNA libraries from the testes of high and low sperm motility groups to evaluate the expression of mRNAs and lncRNAs by Illumina high-throughput sequencing, and we identified critical candidate lncRNAs related to sperm motility. Specifically, we identified 229 DE lncRNAs and 2673 DE mRNAs between high and low sperm motility pigeon testes.

Recently, many studies have found that lncRNAs play critical roles in the regulation of sperm development [8,26,27]. Testis lncRNAs have been identified in humans [9], mice [28], pigs [29], and chickens [30]. This study is the first to report the expression profile of lncRNAs and mRNAs in pigeon testes. We found that lncRNAs identified in this study have shorter transcript lengths and fewer exons than protein coding transcripts, which are features that are similar to those from other species. The characteristics of lncRNAs were consistent with previous studies, indicating that the lncRNA results obtained in this study were reliable. According to the seq-RNA results, the length of 45.46% of lncRNAs was shorter than 1000 bp, while 78.78% of mRNAs were longer than 1000 bp. Interestingly, the average expression level of lncRNAs was higher than that of mRNAs in pigeon testes, which indicated that lncRNAs might have potential functions in sperm motility.

It is well known that a series of exquisitely regulated changes in the expression of many genes during testis development and spermatogenesis may further affect sperm motility. In this study, we identified many DE mRNAs that may be related to sperm motility, including DAZL, ARL3, CDK12, DNAH12 and DNM1. For example, the expression of DAZL is a hallmark of mouse germ cells and is also essential throughout male gametogenesis [31]. Mice lacking DAZL were infertile; when DAZL was knocked out, germ cells showed a defect in the differentiation of Aal spermatogonia to A1 spermatogonia [32]. The upregulation of DAZL expression might be related to the high sperm motility in pigeon. 

It is well established that multiple signaling pathways and regulatory mechanisms are involved in the control of sperm motility [33,34]. As such, GO terms and KEGG pathways were performed to further reveal the biological functions of DE mRNAs and target genes of DE lncRNAs in pigeon sperm motility. We found that these lncRNAs and mRNAs were both involved in the regulation of ATP binding, cell proliferation, hormone activity, oxidation-reduction processes and spermatogenesis. For example, ATP binding has been shown to be involved in the chemo-mechanical transduction of motor proteins in the flagella [35]. The oxidation-reduction reaction has been shown to play an important role in maintaining the normal mechanism of cellular signaling [36]. Some hormones have been shown to be essential for spermatogenesis and sperm maturation [37].

These results suggested that sperm motility could be regulated by the DE lncRNAs and mRNAs found in our results. In addition, we found that CDH1, LOC102096126 (regucalcin) and LOC102087251 (Annexin A8) were enriched in the “calcium ion binding” GO term, which has a key role in spermatogenesis [34]. Moreover, CDH1 was enriched in the cell adhesion molecule pathway. Expression of CDH1 is essential for the survival of germ cells; knockout of CDH1 in germ cells results in a significant reduction or complete elimination of germ cells [38]. Regucalcin is a calcium-binding protein, and transgenic rats overexpressing regucalcin exhibit a protective effect on testicular damage [39]. 

A major function of lncRNAs is to regulate the expression of neighboring protein-coding genes through transcriptional coactivation/repression [40,41]. There might be a mechanism by which lncRNAs could affect the spermatogenesis by mediating the regulation of corresponding target mRNAs. Therefore, in this study, the DE cis-target genes that were located within 100 kb upstream and downstream of the 229 DE lncRNAs were used to predict their potential roles in the regulation of sperm motility. As a result, we found that the DE coding gene UBB may be regulated by the DE lncRNA MSTRG.7787.5. UBB is a gene that modifies molecules through the addition of ubiquitin (Ub), and it is an essential and highly-conserved small protein in eukaryotic cells that is involved in regulating protein degradation, signal transduction, transcriptional regulation and germ cell development [42,43]. Ubiquitination is a posttranslational modification process that plays a key role in regulating spermatogenesis, from the differentiation of spermatogonial stem cells to the formation of mature sperm, and it controls the expression of enzymes and structural proteins in spermatogenesis [44,45,46]. The disruption of UBB leads to male and female infertility resulting from the arrest of germ cells at meiosis prophase 1 [47]. However, it is still unclear how decreased Ub levels due to UBB deficiency result in infertility. Therefore, the UBB gene could be a candidate gene for further study in terms of how it affects sperm motility. Understanding the mechanism by which our screened lncRNAs and mRNAs might have an impact on sperm motility could be useful for animal breeding and is also conducive to the development of animal husbandry.

In conclusion, we obtained the expression profiles of lncRNAs and mRNAs from the testes of pigeons with high sperm motility and low sperm motility, and we used RNA-seq to perform this novel study. A number of DE mRNAs and lncRNAs were found to be associated with sperm motility in pigeons. In addition, DE lncRNAs that were obtained from our data provide new insights that add to the understanding of the regulation of sperm motility in pigeon, which could affect the expression of target genes. The lncRNA MSTRG.7787.5 could play a regulatory role in sperm motility by affecting its potential target gene UBB. Therefore, lncRNA MSTRG.7787.5 could be a candidate lncRNA for regulating sperm motility. However, further studies are needed to validate its function and mechanism in detail.

## Figures and Tables

**Figure 1 genes-11-00349-f001:**
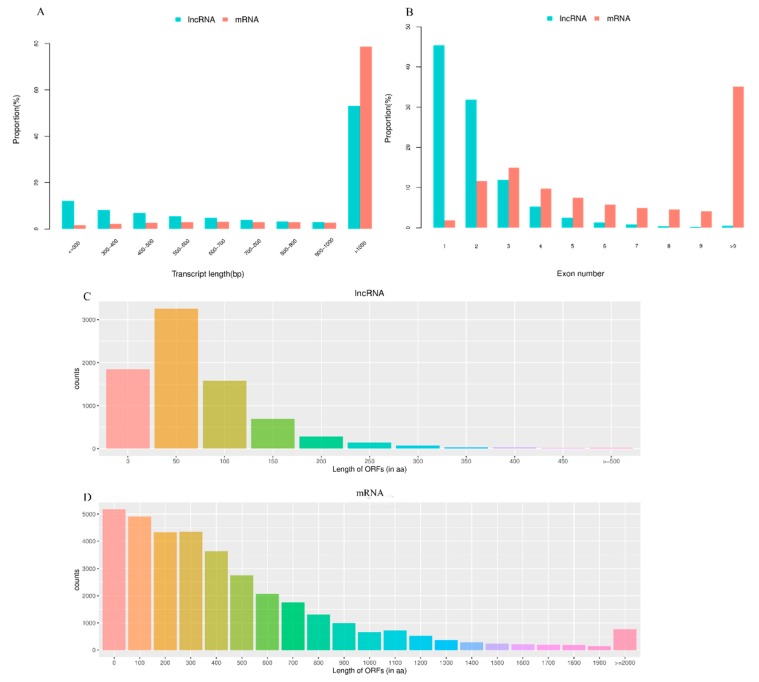
Genomic features of lncRNAs in the testes of pigeon. (**A**) The transcript length distribution of lncRNAs and mRNAs. (**B**) The exon number distribution of lncRNAs and mRNAs. (**C**) The ORFs length distribution of lncRNAs. (**D**) The ORFs length distribution of mRNAs.

**Figure 2 genes-11-00349-f002:**
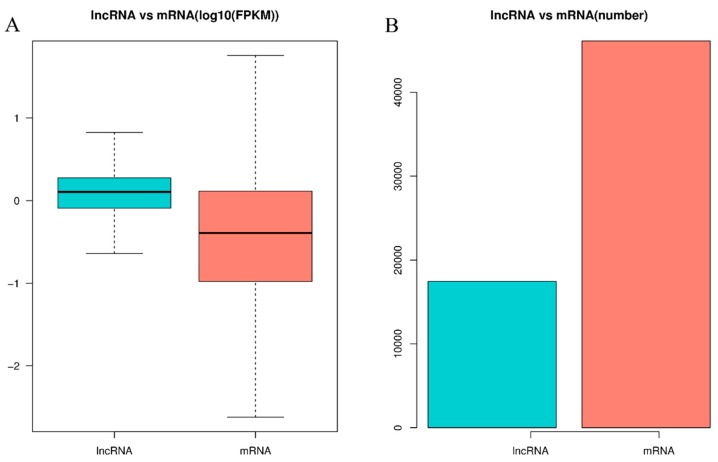
The expression levels and amounts of lncRNAs and mRNAs. (**A**) Boxplots of lncRNAs and mRNAs expression levels (with log10 FPKM method) in the high and low sperm motility groups. (**B**) The number of lncRNAs and mRNAs in testes with high sperm motility.

**Figure 3 genes-11-00349-f003:**
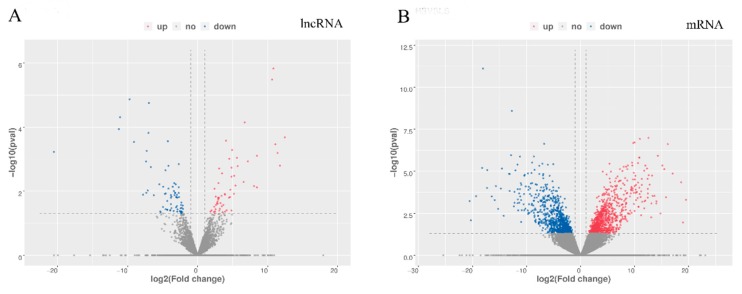
The differential expression of lncRNAs and mRNAs between high and low sperm motility groups. (**A**) Differential expression of lncRNAs. The blue points denote significantly up-regulated lncRNAs, while the red points denote significantly down-regulated lncRNAs in high sperm motility testes. (**B**) Differential expression of mRNAs. The blue points denote significantly up-regulated mRNAs, while the red points denote significantly down-regulated mRNAs in high sperm motility testes.

**Figure 4 genes-11-00349-f004:**
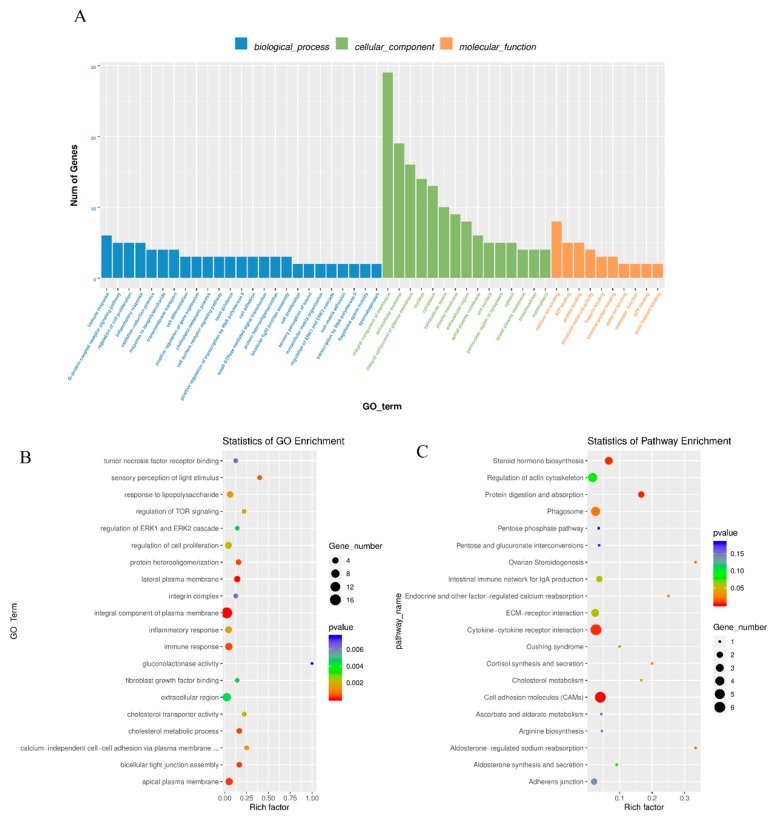
GO and KEGG analysis of differential mRNA expression. (**A**) Histogram of GO enrichment of DE mRNAs. (**B**) Scatter plot of GO enrichment for DE mRNAs. (**C**) Scatter plot of KEGG enrichment for DE mRNAs.

**Figure 5 genes-11-00349-f005:**
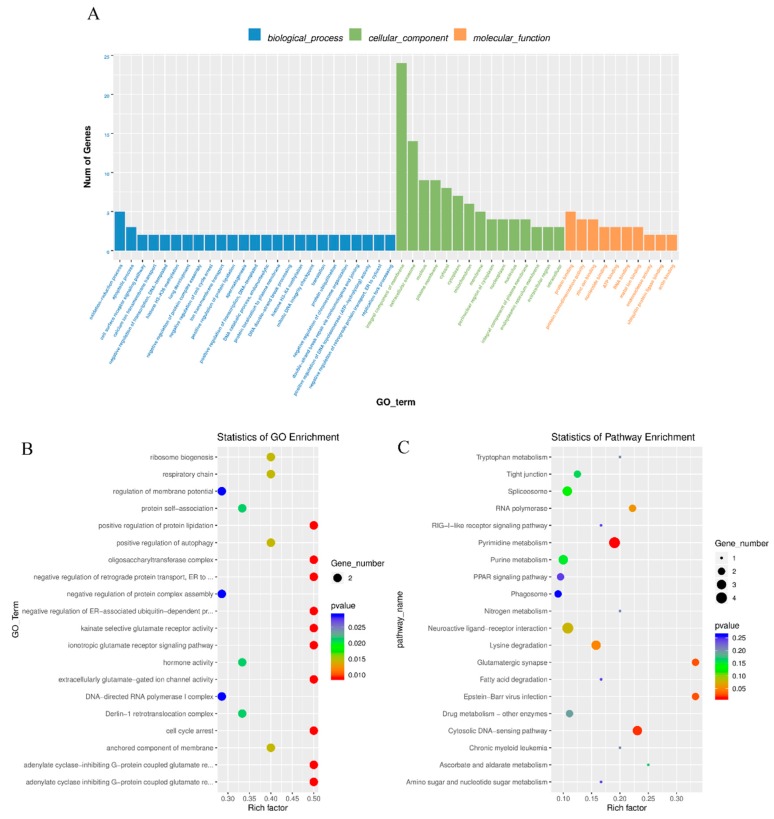
GO and KEGG analysis of differential lncRNA expression. (**A**) Histogram of GO enrichment of DE lncRNAs. (**B**) Scatter plot of GO enrichment for DE lncRNAs. (**C**) Scatter plot of KEGG enrichment for DE lncRNAs.

**Figure 6 genes-11-00349-f006:**
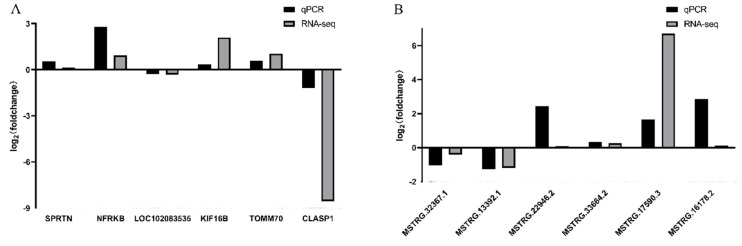
Validation of RNA-seq by qRCR. (**A**) qRCR validation of 6 mRNAs. (**B**) qRCR validation of 6 lncRNAs.

**Table 1 genes-11-00349-t001:** Sperm motility of ten pigeons involved in RNA-seq.

	High Sperm Motility Group						Low Sperm Motility Group					
Sample	HS1	HS2	HS3	HS4	HS5	Mean	LS1	LS2	LS3	LS4	LS5	Mean
Sperm motility	7	8.5	8.3	8.2	9	8.2	1.3	3.5	1	3	3	2.4

**Table 2 genes-11-00349-t002:** Differentially expressed lncRNA-gene pairs between high and low sperm motility groups.

Gene Name	lncRNA Transcript Name	Cis Location (bp)	Pearson Correlation Coefficient
CCSER1	MSTRG.102.1	1k	1
BBS9	MSTRG.18285.3	1K	1
SVIP	MSTRG.2435.3	1K	−0.29
AGR3	MSTRG.25354.1	1K	1
RPL34	MSTRG.33554.2	1K	−0.26
UBB	MSTRG.7787.5	1K	1
SSU72	MSTRG.31781.3	10K	−0.38
SRSF2	MSTRG.9899.4	1K	1

**Table 3 genes-11-00349-t003:** Co-enriched GO terms of DE lncRNA and mRNA.

GO Term	GO Function	p-Value
GO:0031225~anchored component of membrane	cellular_component	0.014
GO:0005179~hormone activity	molecular_function	0.021
GO:0005262~calcium channel activity	molecular_function	0.037
GO:0072686~mitotic spindle	cellular_component	0.040
GO:0005856~cytoskeleton	cellular_component	0.042

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
