# Peer review of "Analysis of Long Noncoding RNA and mRNA Expression Profiles of Testes with High and Low Sperm Motility in Domestic Pigeons (Columba livia)"

_genes, 2020, doi:10.3390/genes11040349_

Round 1
Reviewer 1 Report
The main aim of the manuscript entitles “Analysis of long noncoding RNA and mRNA expression profiles of testes with high and low sperm motility in domestic pigeons (Columba livia)” was to study the testis transcriptome (lncRNA and mRNA) of pigeons with high and low sperm motility. Obtained results showed 2673 mRNAs and 229 lncRNAs significantly differentially expressed between the high and low sperm motility groups. In addition, GO and KEGG annotation analysis showed that target genes of DE lncRNAs and DE mRNAs were related to calcium ion binding, ATP binding, and spermatogenesis. The present work present results useful to increase our knowledge about the gene and pathway involved in spermiogenesis in birds. The following comments would have to be considered.
Lines 67-81. This paragraph can be reduced to the mention of the kit used for the construction of the library.
Lines 147-149 and figure 2. This sentence is not clear, please details with more details, including units.
Include edgeR software in the material and method section.
mRNA Gene Ontology analysis have to be (GO and KEGG) have to be described in the material and method section.
Why did the authors select randomly 6 gene for validate by qPCR instead of select the gene with higher DE?
An statistic method could be include to compare the NGS and qPCR results.
Why did the authors use the a 100kbp upstream and downstream to select coding genes near DE LncRNAs. Please in the, justify or include a reference that that support it.
Include the Pearson Correlation Coefficient material and method section.
Reviewer 2 Report
In this paper the Authors investigated the differential expression of lncRNAs and mRNAs in testes of pigeons showing differential sperm motility. The aim of this study is to get new insights into the molecular mechanisms involved in the sperm motility.
Overall, while the results section is quite clear and easily understandable, the discussion section must be improved to help the reader to understand the novelty and the meaning of the main findings of this study. If possible, a table summarizing the main differences found between the two groups of pigeons in terms of genes or pathways affected should help the reader to keep in mind the main results of this work.
Furthermore, I am wondering if the method used for assessing the sperm motility of pigeons is sufficiently accurate. More details are needed in order to better explain the methodology used.
Some sentences, like that reported in line 49, seem to be incomplete. Also the sentence reported in lines 275-277 should be rewritten to improve its meaning.
There is probably a mistake in line 203 where Table 1, instead of Table 2, is quoted.
Round 2
Reviewer 1 Report
All comments and suggestion did during the first revision were introduced in the new version of the present manuscript.
Reviewer 2 Report
Accept in present form